# Disparities in Coverage of Adult Immunization among Older Adults in India

**DOI:** 10.3390/vaccines10122124

**Published:** 2022-12-12

**Authors:** Damini Singh, Abhinav Sinha, Srikanta Kanungo, Sanghamitra Pati

**Affiliations:** ICMR—Regional Medical Research Centre, Bhubaneswar 751023, India

**Keywords:** vaccines, adult immunization, disparities, VPDs, India, pneumococcal, influenza, hepatitis B, typhoid, diphtheria and tetanus, multimorbidity

## Abstract

A lack of a universal adult immunization scheme in India poses a challenge to achieve universal health coverage. Healthcare disparity is one of the biggest challenges in low- and middle-income countries such as India. We aimed to estimate the disparities in coverage of various adult vaccines among older adults in India using nationally representative data. An observational analysis among 31,464 participants aged ≥60 years from the Longitudinal Ageing Study in India, 2017–2018, was conducted. Vaccination coverage across wealth quintiles and selected non-communicable diseases were reported as frequencies and weighted proportions along with their 95% confidence intervals as a measure of uncertainty. The highest coverage was of the diphtheria and tetanus vaccine (2.75%) followed by typhoid (1.84%), hepatitis B (1.82%), influenza (1.59%), and pneumococcal (0.74%). The most affluent groups had a higher coverage of all vaccines. Participants having high cholesterol, psychiatric conditions, and cancer had the highest coverage of all vaccines. Overall, a very low coverage of all vaccines was observed. The coverage was influenced by social determinants of health, depicting a disparity in accessing immunization. Hence, at-risk groups such as the deprived and multimorbid patients need to be covered under the ambit of free immunization to achieve universal health coverage.

## 1. Introduction

Immunization has been considered one of the most cost-effective public health interventions worldwide [1]. Its efficacy and transparency have already been marked for eradicating smallpox globally. The launch of the Expanded Program of Immunization (EPI) by the World Health Organization (WHO) has configured the Universal Immunisation Program (UIP) which aims at achieving coverage of vaccines for all neonates, children, and pregnant women [1]. However, some vaccine-preventable diseases (VPDs) may equally affect the adult population. Moreover, owing to the current demographic transition in low- and middle-income countries (LMICs) such as India, the adult population is on the rise [1]. An increase in the adult population highlights the urgent need for their immunization to achieve universal health coverage (UHC). Adult immunization has become a major concern, especially in LMICs such as India where this group is more susceptible to acquiring diseases during outbreaks or various other conditions associated with non-communicable diseases (NCDs) [1,2]. Additionally, waning immunity and age-related factors (including immunosenescence) also highlight the need for adult vaccination [2,3].

The measles and rubella (MR) vaccine and consideration of the human papillomavirus (HPV) for potential inclusion in the UIP under the ambit of public vaccination efforts have brought the transition of childhood vaccination programs towards adolescents. Vaccines such as tetanus toxoid (TT) vaccines for pregnant ladies and Japanese encephalitis (JE) vaccines for adults residing in endemic districts are also provided free of cost by the government. Moreover, TT and JE have the highest coverage among adult vaccines [1]. However, other adult vaccines such as influenza, pneumococcal, hepatitis B, typhoid, and diphtheria and tetanus vaccines are also available but are often underutilized. Still, adult vaccination remains a challenge, especially amongst deprived groups [2,4]. Additionally, various groups such as adults with multimorbidity (simultaneous occurrence of two or more chronic conditions in an individual without considering index disease) are at a higher risk of contracting VPDs and may immediately need vaccination [2,3].

Adult immunization is one of the keys to empowering a life course initiative for health care services. The sustainability of vaccines depends on availability and affordability. However, India does not have a clear mandate for providing universal adult vaccination which makes these vaccines underused. This becomes grave with the under-recognition of outbreaks and a deficiency of data on the real burden of VPDs among adults [1,5]. Further, this points toward an urgent need to generate evidence on the present adult vaccination coverage in India. This would help in planning future policies for transitioning conventional childhood and recent adolescent immunization towards adults to avert mortality and morbidity in this age group. Additionally, there is a dearth of literature on adult immunization coverage in India. The available studies represent either a city or some particular region with no national-level data. Hence, this study was conducted to estimate the coverage of the influenza vaccine, pneumococcal vaccine, hepatitis B vaccine, typhoid vaccine, and diphtheria and tetanus (DT) vaccine among adults in India using nationally representative data.

## 2. Methods

An observational analysis based on the first wave of the Longitudinal Ageing Study in India (LASI), 2017–2019, was conducted. The LASI is a community-based study proposed to be conducted every two years among the aging population aged ≥60 years and above and their spouses, irrespective of age. The LASI is a multi-partner undertaking by the Harvard School of Public Health (HSPH), the International Institute of Population Sciences (IIPS), and the University of Southern California. The LASI utilizes a multistage stratified area probability cluster sampling design to achieve the ultimate sampling unit. A response rate of 87.3% was registered by the first wave of the study. Face-to-face interviews were conducted among 31,464 participants aged ≥60 years which formed the ultimate sample for our study. The detailed methods related to the LASI survey can be found on their website [6].

### 2.1. Outcome Variables

The individual survey schedule of the LASI asked “have you ever received any immunizations for adults, such as the influenza vaccine, pneumococcal vaccine, hepatitis B vaccine, or typhoid vaccine?”. Multiple answers were allowed for the above question. If the response was “yes” for a particular vaccine, then we considered an individual to be vaccinated for that particular vaccine such as influenza vaccine, pneumococcal vaccine, hepatitis B vaccine, or typhoid vaccine which formed the main outcomes of interest for our study.

### 2.2. Independent Variables

We took into account the following socio-economic and demographic factors: age (in years), sex, residence, caste, education, occupation, marital status, MPCE quintile, and health insurance. Answers to the age-based question “how old were you on your last birthday?” were categorized into three groups, namely 60–69 years, 70–79 years, and >80 years. Sex was recorded based on observation as male or female. The residence of the respondents was divided as urban or rural. Caste was classified into four groups, namely scheduled caste, scheduled tribe, other backward class, and others (includes: no caste/tribe and none of these) based on two questions “what is your caste or tribe?” followed by “Do you belong to a scheduled caste, a scheduled tribe, other backward class, or none of these?”. Education of the respondents was assessed through the question “have you ever attended school?” with responses yes or no corresponding with formal education and no formal education, respectively. The current occupation status of the respondent grouped as currently employed or currently unemployed was based on the responses to “have you ever worked for at least 3 months during your lifetime?”. Marital status was classified based on “what is your current marital status?” with responses grouped as with partners (currently married or live-in relationship) or without partners (widowed, divorced, separated, deserted, or never married). The economic status was based on the monthly per capita expenditure (MPCE) grouped into quintiles ranging from the most deprived to the most affluent class. The health insurance coverage among participants was assessed through “are you covered by health insurance?” with responses as yes or no. Based on an extensive literature search, ten self-reported most common NCDs such as hypertension, diabetes, cancer, chronic lung disease, chronic heart disease, stroke, arthritis, psychiatric problems, and high cholesterol were taken into account based on one of the questions: “has any health professional ever diagnosed you with the following chronic conditions or diseases?”. Multimorbidity was defined as having two or more chronic conditions out of the above-mentioned chronic conditions.

### 2.3. Statistical Analysis

We employed STATA version 16.0 (Stata Corp™, College Station, TX, USA) for the analysis. Descriptive statistics included mean and standard deviation along with range, frequencies, and percentages to assess the respondents’ background characteristics and estimate the coverage of vaccines. Vaccination coverage across various socio-demographic attributes and selected NCDs was reported. A multi-variable logistic regression model assessed the association between various levels of uptake for adult vaccines and wealth quintiles adjusted for other socio-demographic characteristics. A weighted analysis was conducted to compensate for complex survey designs. We reported a 95% confidence interval (CI) for all weighted proportions as a measure of uncertainty.

### 2.4. Ethics Statement

The LASI received ethical endorsement from the Indian Council of Medical Research (ICMR), New Delhi, and the International Institute for Population Sciences (IIPS), Mumbai. Entrants were given a prospectus containing the information on the aims and objectives of the survey, confidentiality of their personal information, and safety of health assessment. Written consent forms were subjugated at household and individual levels. The LASI employed informed written consent forms. This study is based on anonymous secondary data from the LASI; hence, no ethical concerns arise.

## 3. Results

This study was based on 31,464 participants aged ≥60 years with a mean age of 68.87 ± 7.51 years. Almost half of the participants (58.51%) were 60–69 years of age (Table 1). We observed a female predilection (52.55%) in the study population. Around 70.55 of the respondents lived in rural areas. We found that 43.48% of the participants had formal education and 74.06% of respondents were currently employed. We observed that 18.24% of the participants had health insurance coverage.

The overall coverage of the diphtheria and tetanus vaccine (2.75% (95% CI: 2.75–3.12)) was highest, followed by the typhoid vaccine (1.84% (95% CI: 1.69–1.99)), hepatitis B vaccine (1.82% (95% CI: 1.67–1.97)), influenza vaccine (1.59% (95% CI: 1.45–1.73)), and pneumococcal vaccine (0.74% (95% CI: 0.65–0.84)). Further, it was observed that the participants from the most affluent group had a higher coverage of adult vaccinations compared to any other group. Diphtheria and tetanus vaccines were mostly (4.21%) taken by the most affluent group. The pneumococcal vaccine had the minimum coverage among the most deprived group, i.e., only 0.16% (Table 2).

The adjusted multi-variable model revealed that the most affluent group had a higher chance of getting vaccinated for influenza (AOR: 3.32 (95% CI: 2.52–4.39)), pneumococcal (AOR: 5.53 (3.41–8.99)), hepatitis B (AOR: 5.24 (95% CI: 3.99–6.87)), typhoid (AOR: 3.53 (95% CI: 2.65–4.70)), and diphtheria and tetanus (AOR: 3.60 (95% CI: 2.96–4.39)) than the most deprived group (Table 3).

It was found that respondents with psychiatric problems (5.28%) followed by high cholesterol (3.69%), multimorbidity (2.59%), and stroke (2.46%) had taken influenza vaccines more than participants having other selected chronic conditions. Pneumococcal vaccination coverage was observed to be higher among respondents having high cholesterol (2.56%) followed by psychiatric problems (2.29%) and cancer (2.20%). The coverage for the hepatitis B vaccine was found to be higher in respondents with high cholesterol (5.53%) followed by cancer (5.51%) and psychiatric problems (5.06%). Typhoid vaccination coverage was found to be higher in respondents having high cholesterol (4.59%) followed by psychiatric problems (5%) and cancer (4.46%). Respondents with high cholesterol (10.05%) followed by psychiatric problems (6.86%) and cancer (6.84%) had higher coverage of diphtheria and tetanus vaccines (Table 4).

## 4. Discussion

The overall coverage of adult vaccination was considerably low among the participants belonging to deprived groups. The highest coverage was of the DT vaccine followed by those of typhoid, hepatitis B, influenza, and pneumococcal. Participants having high cholesterol, psychiatric conditions, and cancer had the highest coverage for all vaccines.

We observed that the DT vaccine had the highest coverage followed by typhoid, hepatitis B, influenza, and pneumococcal vaccines. A recent facility-based study conducted at an adult vaccination center in Jodhpur observed that tetanus toxoid, anti-rabies, and yellow fever vaccines had the highest coverage [1]. However, the coverage of the hepatitis B vaccine (8%), followed by the pneumococcal vaccine (7%) and typhoid vaccine (3%) was reported higher than the findings of our study [1]. Interestingly, the coverage of the influenza vaccine (1%) was found to be lower compared to the present study. Notably, there is a dearth of literature on adult vaccination in India which makes comparing our findings with similar studies difficult. A 2018 US report on adult vaccination surveillance observed the coverage of the influenza vaccine to be around 46.1%, hepatitis B around 30%, and pneumococcal around 23.3%, which is significantly lower than the coverage of adult immunization in India [7]. The major reason for this could be the disparity in accessing vaccines in India, as adult vaccination is not covered in the routine universal immunization schedule.

The increase in antibiotic-resistant bacterial strains such as *S. pneumonia* [8] due to over-the-counter drugs has led to a rise in pneumococcal infections which may also invade the bloodstream, causing meningitis. Older adults are particularly at a higher risk of becoming severely ill and dying; hence, they must be vaccinated [8,9]. This could be the probable reason for the higher coverage of the pneumococcal vaccine among participants aged ≥61 years. Influenza caused by the influenza virus affects individuals of all ages but it has the highest risk of complications among older adults [8]. However, the effectiveness of the influenza vaccine is lower among older adults [10,11]. The WHO advises for an annual influenza immunization for older adults [8,12]. Our findings are consistent with the WHO’s recommendations for vaccinating older adults; however, the coverage is considerably low which may pose a challenge for UHC [8]. Pneumococcal and influenza vaccines are indicated among diabetes patients since they have irregularities in immune function, leading to a rise in morbidity and mortality from infection [11,13]. Further, diabetics have a higher chance of complications from influenza and pneumococcal infections leading to hospitalization and death [11]. Diabetics have an appropriate humoral immune response to immunization [11]. Nonetheless, previous studies have reported that the influenza vaccine has reduced hospital admission during epidemics, whereas the pneumococcal vaccine has been effective in reducing bacteremic infections [11]. Our findings show a very low coverage of both of these vaccines among diabetics which is a grave concern.

Typhoid fever continues to be an endemic disease in Southeast Asia with a substantial number of cases among teenagers and young adults [14]. Poor sanitation facilities, especially among deprived groups, is a major cause of typhoid [14,15]. However, we observed that deprived strata had a lower coverage of the typhoid vaccine which may lead to an increased case burden in this group. Additionally, both acute and chronic infections of hepatitis B cause disproportionately higher mortality and morbidity in LMICs, where it is a significant public health issue [16]. Evidence suggests catch-up immunization for younger adults is beneficial above costs [17]. Hence, for adults in India, catch-up immunization must be planned for those who were not vaccinated in their childhood. This should specifically be for the adults who are at a higher risk of infection such as drug abusers and individuals with liver diseases. Similar to the hepatitis B vaccine, the coverage of the DT vaccine during childhood is high but previous studies have reported unsatisfactory antibody levels among adults [18]. This highlights a need for adult DT vaccination [19]. It is to be noted that we found a low coverage of all vaccines which might lead to a high disease burden among adults.

We observed a variation in the coverage of various adult vaccines across wealth indexes. Participants belonging to the most affluent groups had the highest coverage of all vaccines. Our findings are consistent with the reports from other LMICs such as China, where a study observed that people living with a finance-reimbursed vaccination policy had a higher vaccination rate [20]. Moreover, a study conducted in Pakistan observed that the majority of the participants were not receiving adult vaccines due to lack of awareness [21]. A probable reason for this could be their ability to pay. Since we do not have a universal program for adult vaccination in India, individuals need to pay to receive the vaccines. However, the disparities across deprived and affluent groups may lead to a low coverage of vaccines which needs to be equitably dealt with. These findings are relevant with the conceptual framework of the Commission on Social Determinants of Health (CSDH) [22,23]. Additionally, our findings are consistent with the findings of a systematic review which investigated the role of social determinants and seasonal influenza vaccination in adults aged 65 years and above and found that age, gender, education, ethnicity, etc. influenced immunization [22]. Here, it is worth noting that older adults in India might need information, education, and communication (IEC) to take up vaccination as, conventionally, it is thought to be for children. Lack of awareness can be a major barrier in increasing immunization coverage which needs to be strengthened.

### 4.1. Implications for Policy and Practice

The National Technical Advisory Group on Immunization in India (NTAGI) does not provide a clear mandate on adult vaccination in India. However, their recommendations can shape the future course of adult immunization in India. Similar to COVID-19 vaccination, a phase-wise coverage based on the assessment of risk factors is required for all adult vaccines in India. Additionally, the provision of subsidized vaccines can also help in achieving higher immunization coverage. Along with the at-risk groups, women and economically deprived groups also need to be focused on. People living in hard-to-reach areas and tribal groups also are vulnerable to VPDs; hence, they require support for vaccination. The Ayushman Bharat scheme should establish adult vaccination in the bundle of services for the deprived class. Systematic mechanisms to vaccinate individuals with chronic conditions and multimorbidity is required. For equitable and egalitarian access, the availability of vaccines should be at the nearest healthcare centers. Furthermore, IEC and behavioral change communication (BCC) are required for beneficiaries to understand the need for vaccines. Adult immunization should be included in mainstream medical education and training curricula. Future studies on operational feasibility and enablers and barriers to adult immunization need to be explored.

### 4.2. Strengths and Limitations

This study used nationally representative data to investigate adult immunization coverage in India. To the best of our knowledge, this is the first study on nationwide adult immunization coverage. However, our study is limited by self-reported vaccination status which is susceptible to recall bias.

## 5. Conclusions

We observed a very low coverage of all vaccines among adults. Furthermore, the coverage was higher in affluent groups, depicting a disparity in accessing immunization. However, universal vaccination may not be feasible in India due to the huge population of at-risk groups and disadvantaged sections of society such as deprived strata and women, who need to be covered under the ambit of free immunization, which can help in achieving universal health coverage.

## Figures and Tables

**Table 1 vaccines-10-02124-t001:** Characteristics of the study population.

Characteristics	Weighted n (%)
Age in years (n = 31,464)	
60–69	18,410, (58.51)
70–79	9501, (30.20)
>80	3553 (11.29)
Sex (n = 31,464)	
Male	14,931, (47.45)
Female	16,533, (52.55)
Residence (n = 31,464)	
Rural	22,196, (70.55)
Urban	9268, (29.45)
Caste (n = 31,198)	
Scheduled caste	5926, (18.99)
Scheduled tribe	2546, (8.16)
Other backward class	14,175, (45.44)
Others	8551, (27.41)
Education (n = 31,464)	
Formal Education	13,681, (43.48)
No formal education	17,783, (56.52)
Occupation (n = 31,460)	
Currently Employed	23,151, (74.06)
Currently Unemployed	8309, (26.41)
Marital Status (n = 31,464)	
Without partner	11,928, (37.91)
With partner	19,536, (62.09)
MPCE Quintile (n = 31,464)	
Most deprived	6829, (21.70)
2	6832, (21.71)
3	6590, (20.95)
4	6038, (19.19)
Most affluent	5175, (16.45)
Health insurance (N = 31,162)	
Yes	5685, (18.24)
No	25,477, (81.76)

**Table 2 vaccines-10-02124-t002:** Coverage of various adult vaccines across wealth quintiles among older adults in India.

Wealth Quintiles	Influenza Vaccine n, % (95% CI)	Pneumococcal Vaccine n, % (95% CI)	Hepatitis B n, % (95% CI)	Typhoid n, % (95% CI)	Diphtheria and Tetanus n, % (95% CI)
Most deprived	57, 0.84, (0.64–1.08)	11, 0.16, (0.08–0.28)	37, 0.55, (0.38–0.75)	56, 0.82, (0.62–1.07)	95, 1.40, (1.13–1.71)
2	78, 1.16, (0.91–1.43)	34, 0.51, (0.35–0.70)	101, 1.49, (1.21–1.81)	103, 1.52, (1.24–1.84)	190, 2.81, (2.42–3.22)
3	90, 1.41, (1.13–1.72)	40, 0.63, (0.44–0.85)	102, 1.60, (1.30–1.93)	124, 1.93, (1.61–2.30)	190, 2.97, (2.56–3.41)
4	94, 1.58, (1.27–1.93)	59, 0.99, (0.75–1.27)	138, 2.32, (1.95–2.73)	118, 1.98, (1.64–2.37)	220, 3.70, (3.23–4.20)
Most affluent	172, 3.37, (2.88–3.89)	85, 1.68, (1.33–2.05)	185, 3.63, (3.12–4.17)	170, 3.33, (2.85–3.85)	215, 4.21, (3.67–4.79)

**Table 3 vaccines-10-02124-t003:** Association between uptake of various adult vaccines and wealth quintiles among older adults in India.

Wealth Quintiles	Influenza Vaccine AOR (95% CI)	Pneumococcal Vaccine AOR (95% CI)	Hepatitis B AOR (95% CI)	Typhoid AOR (95% CI)	Diphtheria and Tetanus AOR (95% CI)
Most deprived	Ref				
2	1.25	1.99	1.86	1.90	1.59
	(90.98–1.71)	(1.16–3.41)	(1.37–2.52)	(1.40–2.59)	(1.28–1.98)
3	1.64	2.88	2.18	2.24	2.01
	(1.21–2.21)	(1.73–4.80)	(1.62–2.93)	(1.66–3.02)	(1.63–2.47)
4	2.26	3.61	3.38	2.41	2.67
	(1.69–3.02)	(2.18–5.95)	(2.55–4.48)	(1.79–3.25)	(2.18–3.27)
Most affluent	3.32	5.53	5.24	3.53	3.60
	(2.52–4.39)	(3.41–8.99)	(3.99–6.87)	(2.65–4.70)	(2.96–4.39)

Adjusted for age, sex, residence, caste, occupation, and health insurance.

**Table 4 vaccines-10-02124-t004:** Coverage of adult vaccines across selected non-communicable diseases among adults in India.

Non-Communicable Disease	Influenza Vaccine n, % (95% CI)	Pneumococcal Vaccine n, % (95% CI)	Hepatitis B Vaccine n, % (95% CI)	Typhoid Vaccine n, % (95% CI)	Diphtheria and Tetanus n, % (95% CI)
Hypertension	334, 2.28	139, 0.95	360, 2.46	403, 2.75	545, 3.72
	(2.04–2.53)	(0.79–1.11)	(2.21–2.72)	(2.50–3.03)	(3.42–4.04)
Diabetes	145, 2.16	60, 0.90	162, 2.41	138, 2.06	221, 3.29
	(1.82–2.54)	(0.68- 1.15)	(2.06–2.81)	(1.73–2.42)	(2.87–3.75)
Cancer	5, 1.37	7, 2.20	18, 5.51	15, 4.46	23, 6.84
	(0.49–3.50)	(0.85–4.32)	(3.26–8.48)	(2.56–7.38)	(4.47–10.27)
Chronic lung disease	86, 2.31	40, 1.09	68, 1.83	90, 2.43	131, 3.54
	(1.85–2.85)	(0.77–1.46)	(1.42–2.32)	(1.95–2.97)	(2.96–4.17)
Chronic heart disease	51, 2.41	32, 1.53	53, 2.50	52, 2.47	76, 3.60
	(1.79–3.14)	(1.03–2.11)	(1.87–3.24)	(1.83–3.19)	(2.82–4.45)
Stroke	27, 2.46	14, 1.29	28, 2.52	32, 2.84	51, 4.59
	(1.59–3.49)	(0.68–2.09)	(1.67–3.59)	(1.96–4.01)	(3.41–5.95)
Arthritis	207, 2.37	78, 0.89	234, 2.68	253, 2.89	289, 3.29
	(2.05–2.70)	(0.70–1.10)	(2.34–3.03)	(2.54–3.26)	(2.93–3.69)
Psychiatric problem	64, 5.28	28, 2.29	61, 5.06	60, 5.00	83, 6.86
	(4.09–6.69)	(1.54–3.32)	(3.87–6.42)	(3.79–6.32)	(5.49–8.42)
High cholesterol	44, 3.69	31, 2.56	67, 5.53	55, 4.59	121, 10.05
	(2.65–4.85)	(1.74–3.62)	(4.32–6.98)	(3.45–5.88)	(8.37–11.84)
Multimorbidity	193, 2.59	94, 1.26	211, 2.83	212, 2.85	318, 4.26
	(2.24–2.97)	(1.01–1.54)	(2.46–3.23)	(2.47–3.24)	(3.82–4.75)

## Data Availability

The dataset analyzed during the current study is available in the LASI data repository held at ICT, IIPS https://iipsindia.ac.in/content/lasi-wave-I (accessed on 10 December 2022). Requests to access the data should be made to datacenter@ipsindia.ac.in.

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
