# Peer review of "Disparities in Coverage of Adult Immunization among Older Adults in India"

_vaccines, 2022, doi:10.3390/vaccines10122124_

Round 1

Reviewer 1 Report

This is very interesting work. The only suggestion is to continue studies.

Author Response

Thank you so much for your valuable time and encouragement. We will surely continue the studies in the domain.

Reviewer 2 Report

This paper reports statistical analyses of disparities in coverage of various vaccines among adults in India using nationally representative data. Overall, the paper is relatively well written, and the statistical analyses follow standard protocols for analyses of sample survey data. Here are some items to attend to in a revision.

First, all of the tables need headings/titles that describe the contents of the tables.

Second, all of the tables also need a full paragraph in the text the describes the contents of the tables and their substantive meanings.

Third, it would be very good to a paragraph in the Conclusion section that reports on findings of disparities of vaccination coverage among other LMICs in other studies. This can be done by conducting a literature review of published studies and then describing any differences and similarities between findings in those studies and those that you find in the present study. You also can comment on remaining research questions and issues that need to be addressed in future studies. This will be quite useful to readers of your article.

Author Response

This paper reports statistical analyses of disparities in coverage of various vaccines among adults in India using nationally representative data. Overall, the paper is relatively well written, and the statistical analyses follow standard protocols for analyses of sample survey data. Here are some items to attend to in a revision.

First, all of the tables need headings/titles that describe the contents of the tables.

Author’s response: We thank the reviewer for pointing this out. We have now added titles for all the tables.

Second, all of the tables also need a full paragraph in the text describes the contents of the tables and their substantive meanings.

Author’s response: We have now added paragraphs in the result section depicting the content of all the tables.

Third, it would be very good to a paragraph in the Conclusion section that reports on findings of disparities of vaccination coverage among other LMICs in other studies. This can be done by conducting a literature review of published studies and then describing any differences and similarities between findings in those studies and those that you find in the present study. You also can comment on remaining research questions and issues that need to be addressed in future studies. This will be quite useful to readers of your article.

Author’s response: We have added a specific conclusion and discussed on similarities and differences between the findings from existing literature. Additionally, as suggested by the reviewer, we have added future research questions in the implications for policy and practice section.

We thank the reviewer for their valuable time and suggestion which have helped us improve the manuscript.

Reviewer 3 Report

This is an interesting article reporting on the disparities in coverage of adult immunization among older adults in India. The authors aimed to estimate the disparities in coverage of various vaccines among adults in India using nationally representative data. An observational analysis among 31,464 participants aged ≥60 years from Longitudinal Ageing Study in India, 2017-18 was done. Vaccination coverage across socio-demographic attributes and selected non-communicable diseases were reported as frequencies and weighted proportions along with their 95% confidence interval as a measure of uncertainty.

The coverage appeared influenced by social determinants of health depicting a disparity in accessing immunization. Hence, at risk groups along with disadvantaged sections such as deprived group and people belonging to most deprived sections of society need to be covered under the umbrella of free immunization to achieve universal health coverage.

Introduction is exhaustive.

Materials & Methods are detailed and clear.

Statistics is appropriate.

Results are well presented and supported by maps, graphs and SI.

Discussion is clear and focuses on the need of reaching a universal vaccination coverage against VPDs to guarantee adequate protection to all age classes.

References are appropriate. 

Minor changes are indicated in the text.

Author Response

Both minor changes: “Titles for tables” have been added now.

We thank the reviewer for giving their valuable time and encouragement.

Round 2

Reviewer 2 Report

The revisions to this manuscript have been responsive and it has been improved accordingly. I have not further suggestions for revision.